# The Nine-Item Internet Gaming Disorder Scale (IGDS9-SF): Its Psychometric Properties among Sri Lankan Students and Measurement Invariance across Sri Lanka, Turkey, Australia, and the USA

**DOI:** 10.3390/healthcare10030490

**Published:** 2022-03-07

**Authors:** Amira Mohammed Ali, Rasmieh Al-Amer, Maha Atout, Tazeen Saeed Ali, Ayman M. Hamdan Mansour, Haitham Khatatbeh, Abdulmajeed A. Alkhamees, Amin Omar Hendawy

**Affiliations:** 1Department of Psychiatric Nursing and Mental Health, Faculty of Nursing, Alexandria University, Smouha, Alexandria 21527, Egypt; mercy.ofheaven2000@gmail.com; 2Faculty of Nursing, Isra University, Amman 11953, Jordan; r.al-amer@outlook.com; 3School of Nursing and Midwifery, Western Sydney University, Penrith, NSW 2751, Australia; 4School of Nursing, Philadelphia University, Amman 19392, Jordan; m.atout@philadelphia.edu.jo; 5School of Nursing and Midwifery, Aga Khan University, Karachi 3500, Pakistan; tazeen.ali@aku.edu; 6Department of Psychiatric and Mental Health Nursing, School of Nursing, The University of Jordan, Amman 11942, Jordan; a.mansour@ju.edu.jo; 7Faculty of Health Sciences, Doctoral School of Health Sciences, University of Pécs, 7621 Pécs, Hungary; khatatbeh.haitham@etk.pte.hu; 8Department of Medicine, Unayzah College of Medicine and Medical Sciences, Qassim University, Unayzah, Al Qassim 52571, Saudi Arabia; 9Department of Biological Production, Tokyo University of Agriculture and Technology, Tokyo 183-8509, Japan; amin.hendawy@gmail.com; 10Department of Animal and Poultry Production, Faculty of Agriculture, Damanhour University, Damanhour 22516, Egypt

**Keywords:** coronavirus disease 2019/COVID-19, Internet Gaming Disorder Scale 9—Short Form (IGDS9-SF), university students, factorial structure/psychometric properties/structural validity/validation, cultur*/collectivisti*/individualis*, invariance, gender, game type

## Abstract

The prevalence of internet gaming disorders (IGD) is considerably high among youth, especially with the social isolation imposed by the ongoing COVID-19 pandemic. IGD adversely affects mental health, quality of life, and academic performance. The Internet Gaming Disorder Scale (IGDS9-SF) is designed to detect IGD according to DSM-IV diagnostic criteria. However, inconsistent results are reported on its capacity to diagnose IGD evenly across different cultures. To ensure the suitability of the IGDS9-SF as a global measure of IGD, this study examined the psychometric properties of the IGDS9-SF in a sample of Sri Lankan university students (N = 322, mean age = 17.2 ± 0.6, range = 16–18 years, 56.5% males) and evaluated its measurement invariance across samples from Sri Lanka, Turkey, Australia, and the USA. Among Sri Lankan students, a unidimensional structure expressed good fit, invariance across different groups (e.g., gender, ethnicity, and income), adequate criterion validity (strong correlation with motives of internet gaming, daily gaming duration, and sleep quality), and good reliability (alpha = 0.81). Males and online multiplayers expressed higher IGD levels, greater time spent gaming, and more endorsement of gaming motives (e.g., Social and Coping) than females and offline players. Across countries, the IGDS9-SF was invariant at the configural, metric, and scalar levels, although strict invariance was not maintained. The lowest and highest IGD levels were reported among Turkish and American respondents, respectively. In conclusion, the IGDS9-SF can be reliably used to measure IGD among Sri Lankan youth. Because the scale holds scalar invariance across countries, its scores can be used to compare IGD levels in the studied countries.

## 1. Introduction

Internet gaming is commonly used as a recreational activity among children, adolescents, and young adults [1]. With the expansion of internet technology, internet gaming has been employed to serve educational purposes, promote physical activity, develop cognitive skills and for therapeutic actions (e.g., reasoning, spatial awareness, and problem-solving) [2]. While most industries were negatively affected during the coronavirus disease 2019 (COVID-19) pandemic, the gaming industry has considerably flourished worldwide, with increased gaming time spent by previous users, numbers of new users, female users, and traffic in online mobile gaming. Indeed, the World Health Organization’s (WHO) collaborative campaign (#PlayApartTogether) has been employed to promote online gaming as a method of fostering socialization while maintaining spatial distancing to prevent infection spread [2,3,4]. Increased risk for excessive gaming and increased screen use time among youth during COVID-19 may be a method to compensate for negative emotions associated with social isolation/being homebound, lack of meaningful activities (e.g., due to closure of schools and workplaces), inability to participate in entertainment activities that were available before the pandemic (e.g., clubs and cinemas), and COVID-19 burnout [3,4,5,6].

Children and adolescents are particularly vulnerable to developing maladaptive patterns of excessive or problematic gaming [3]. Internet Gaming Disorder (IGD) is defined as a behavioral pattern of persistent and recurrent involvement in online and offline games, resulting in remarkable distress and impairment in essential life activities (work and study) for a period of 12 months or more [7,8]. The global prevalence of IGD ranges between 0.7% and 15.6% [9,10]. In a Chinese study comprising 2863 school children, 83.0% played video games during the COVID-19 pandemic. Excessive and pathological gaming were evident in 20.9% and 5.3% of the participants [11].

Factors associated with increased IGD during COVID-19 include male gender, young age, loneliness, lack of parental support and/or supervision, mental health problems, and low socio-economic status [1,3,11,12]. Internet gaming among youth is associated with a plethora of negative consequences: poor health-related quality of life, sleep disturbances, impaired life skills, low self-esteem, concentration problems, poor communication skills, higher social distress, poor real-life relationships, loneliness, aggression, poor academic/work performance, poor impulse control, and tendency toward psychopathology [2,13,14,15,16]. Persistent or excessive gaming may increase the risk of more serious mental health problems [4]. Suicide has been reported among adolescents and emerging adults with psychological predispositions who played videogames that take up many hours a day such as PlayerUnknown’s Battlegrounds (PUBG) [12].

The Diagnostic and Statistical Manual of Mental Disorders (DSM–5) classifies IGD in Section III—disorders requiring further investigation [8,14]. According to the DSM-5, meeting five out of nine normative symptomatic criteria is sufficient for establishing IGD as a diagnosis: (1) preoccupation with playing on the internet/digital games; (2) withdrawal symptoms when internet games are not available; (3) tolerance noted by increased time spent in gaming; (4) relapse noted by failed attempts to quit gaming; (5) loss of interest in other previous hobbies/entertainment behaviors because of, and with the exception of, online games; (6) continued and excessive use of online games despite knowledge of the psychosocial problems it causes; (7) deception of relatives, therapists, or other people about the amount of time spent in gaming; (8) mood modification is noted by use of online games to escape or mitigate negative emotions; and, (9) losing significant interpersonal relationships, work and educational or professional opportunities as a result of participating in internet gaming [7,9]. Because research on IGD is relatively recent, these nine criteria have been described based on existing research on pathological gambling and substance use disorder. Therefore, IGD measurement may be associated with methodological issues i.e., in relation to the definition and presentation of IGD [8]. Among 18 measures designed to assess IGD, the Internet Gaming Disorder Scale 9–Short-Form (IGDS9-SF), a brief form of the IGDS, has been designed to detect all nine diagnostic criteria of IGD [8,16].

The IGDS9-SF has been translated into many languages, including Italian and Albanian [8], Korean [15], Spanish [14], Polish [17], Turkish [10], Arabic [18], and Portuguese [19]. Although numerous studies report good reliability and validity of the IGDS9-SF in samples from Western and developed countries, its psychometric evaluation in various cultural contexts is limited, which may restrict its use as a global measure of IGD [8,17].

Cultural orientations may considerably influence the way through which individuals respond to a symptom scale. For instance, collectivistic cultures put a great appreciation for group values and norms while conditions entailing deviation from group norms (e.g., mental disorders such as IGD) can be stigmatized [20]. Therefore, collectivistic individuals tend to express IGD scores close to the mean, resulting in a minimal range of IGDS9-SF item responses. On the other hand, individualistic cultures entail appreciation of individual goals, values, competition and achievement as a base of social hierarchies. Therefore, individualistic persons tend to compete in gaming in order to achieve higher rankings [8,21].

Measurement invariance is frequently tested to ensure the usability of a measure for comparing the levels of a latent construct across different groups (e.g., cultures, ethnicities, genders, and age) [22,23]. Measurement invariance is assessed at four levels: configural, metric, scalar, and strict. Ideally, an invariant scale successfully reflects similar conceptualization of the underlying latent structure among groups (usually by reporting the same observed scores), similar degree of endorsement of items, and a capacity to objectively compare scale mean scores among groups. Comparisons based on non-invariant measures, especially at the scalar level, are likely to be invalid because group scores are confounded by differences in scaling properties across groups [8,22]. Although all previous findings support the unidimensional structure of the IGDS9-SF, some degree of measurement non-invariance has been expressed, especially in studies comprising English-speaking countries (e.g., Australia, United Kingdome (UK), and the United States of America (USA)) and non-English-speaking samples e.g., Polish and Indian [17,21]. Even among English-speakers from the USA and Australia, investigations of time invariance (three months) uncovered partial metric and scalar non-invariance among Australian gamers [24]. 

Given the widespread incidence of IGD and related mental health and academic adverse effects among youth, careful identification and proper management of IGD in this group may have implications for preventing/mitigating psychiatric comorbidities [23,25]. Lack of validation of IGD measures in developing countries such as Sri Lanka represents a challenge for IGD detection and treatment. It is not clear if IGD levels among youth from Sri Lanka can be compared with IGD levels among countries with evolving economies such as Turkey and developed countries such as Australia and the USA. These four countries do not only vary according to their economy but also according to the dominant cultural orientations. To fill the gap, the current study aimed to examine the structure, invariance, and criterion validity of the IGD9-SF among university students from Sri Lanka. We hypothesized that the IGD9-SF would express a unidimensional structure that would be invariant across different groups (e.g., gender, ethnicity, and income). We also expected that the IGD9-SF would strongly correlate with average daily gaming time, different motives of gaming, as well as sleep quality and quantity. In addition, we expected that the IGD9-SF would express some none-invariance across countries (Sri Lanka, Turkey, Australia, and the USA).

## 2. Materials and Methods

### 2.1. Study Design, Participants, and Procedure

This cross-sectional study is a secondary analysis based on three publicly accessible datasets. The first dataset is affiliated with University of Colombo Faculty of Medicine, Sri Lanka [26] and it is associated with a published study [27]. This dataset comprises a sample of advanced level university students obtained through random cluster sampling from four schools of the Colombo Educational Zone in Sri Lanka. This sample was used to examine the psychometric properties of the IGDS9-SF in Sri Lanka [27]. The sociodemographic, academic, and gaming characteristics of this sample are shown in Table 1.

Invariance of the IGDS9-SF across countries is based on data of the IGDS9-SF only from an international sample, which was integrated from three publicly accessible datasets, including the current sample from Sri Lanka [26]. It also included a sample from Turkey comprising 244 university students who reported playing digital games. Males and females were almost equally represented (N =113, 46.3%) and (N = 131, 53.7%), respectively [28]. No further details are available about the characteristics of those students. The third dataset comprised online gamers from Australia (N = 738, mean age = 25.8 ± 7.6, range = 18–72 years, 49.3% females, 71.7% employed, 32.7% students) and the USA (N = 222, mean age = 27.0 ± 8.0, range = 18–63 years, 54.1% females, 70.3% employed, 39.2% students) [29]. Data collection was obtained through an anonymous online survey conducted through SurveyGizmo, and an ethical approval for data collection was issued by the ethics committee of Cairnmillar Institute. Further details on the characteristics of the participants from those countries are reported in detail elsewhere [30]. Because all the datasets are shared under the terms of creative common license (CC BY 4.0) [26,28] or are in the public domain [29], we did not obtain an ethical approval for the current study.

### 2.2. Data Collection Measures

The questionnaire addressed to students from Sri Lanka included a personal information form inquiring about students’ age, gender, academic major, ethnicity, and gaming experience (hours, type of gaming, age of start). It also comprised other measures, including the Internet Gaming Disorder Scale 9—Short Form (IGDS9-SF), a brief measure of the severity of IGD symptoms [31]. The scale consists of nine items. Each item is rated on a 5-point Likert scale ranging from (1 = never) to (5 = very often). The maximum and minimum scores of the IGDS9-SF are 9 and 45. Higher scores reflect higher levels of problematic internet gaming [31]. The IGDS9-SF was administered both in Sinhala and in English. Its reliability in this Sri Lankan sample is good (coefficient alpha = 0.81).

Motives for Online Gaming Questionnaire (MOGQ), a scale that comprises 27 items, which measure seven major gaming motives: Social (building and maintaining social relationships), Escape (escaping from reality), Coping (coping with stress and distress), Competition (challenging and competing with others), Skill Development (attention and coordination), Fantasy (in-game identities and experience), and Recreation (entertainment and enjoyment). Items are rated on a five-point Likert scale (1 = almost never/never) to (5 = almost always/always) [32]. Its reliability in the current sample is excellent (coefficient alpha = 0.92).

The Single Item Sleep Quality Scale (SISQ) was used to evaluate sleep quality. The response is rated on a scale from 1 to 10, with higher scores indicating better sleep quality [33].

Self-esteem has been assessed by a single question prompting the participants to rate their self-esteem on a 5-point Likert (1 = very poor) to (5 = excellent).

### 2.3. Statistical Analysis

Checking the original Sri Lankan dataset, which comprised 395 responses, for missing data revealed that seventy-three responses had missing data on the IGDS9-SF. Therefore, they were excluded from the analysis, ending with a final sample of 322 respondents—response rate = 82%. The distribution of the IGDS9-SF and MOGQ was examined using Shapiro-Wilk test. Quantitative variables with normal distribution were described by mean and standard deviation while those with non-normal distribution were described using median and interquartile range (IQR: 25–75%). Categorical variables were described using number and percentage.

Based on the literature, the unidimensional structure of the IGDS9-SF was examined in the Sri Lankan sample by confirmatory factor analysis (CFA) using the maximum likelihood method of estimation with bootstrap that generates 2000 random replications. Multigroup CFA was conducted to examine invariance of the IGDS9-SF across countries (Sri Lanka, Turkey, Australia, and the USA) in the international sample as well as across groups of gender, ethnicity, and language used to complete the questionnaire, income, device, game type, and academic major in the Sri Lankan sample (groups are shown in Table 1).

The chi square (χ^2^) index is largely dependent on sample size, and well-fitting models with minor misspecifications may be disqualified based on a significant χ^2^ [25]. Meanwhile, absolute fit indices represent more reliable indicators of model fit because they are sample-size independent. Therefore, we considered model fit in CFA/multigroup CFA to be good or acceptable based on a Comparative Fit Index (CFI) and Tucker–Lewis Index (TLI) equal to or above 0.95 and 0.90, respectively, along with a root mean square error of approximation (RMSEA) and standardized root-mean-square residual (SRMR) less than 0.06 and 0.08, respectively [34].

The internal consistency of the IGDS9-SF was examined by coefficient alpha, alpha-if-item deleted, and item-total correlations. Its criterion validity was evaluated by Spearman’s r correlations with the MOGQ, number of sleeping hours, SISQ, and single-item measure of self-esteem in the Sri Lankan sample. Additionally, a Mann Whitney U test and Kruskal Wallis test were used to examine differences in the number of gaming hours as well as key constructs measured by MOGQ among groups of gender and game type since differences in IGD were depicted in these groups, and they expressed a tendency toward non-invariance in multigroup CFA. Statistical analyses were conducted in SPSS and Amos, and significance was considered at a probability level less than 0.05 in two-tailed tests.

## 3. Results

### 3.1. Confirmatory Factor Analysis and Invariance Analysis of the Internet Gaming Disorder Scale 9—Short Form 

As shown in Table 2, the IGDS9-SF expressed excellent fit among students from Sri Lanka. According to Appendix A, the scale expressed invariance at the configural, metric, and scalar levels among Sri Lankan students across groups of gender, ethnicity, language used to complete the questionnaire, income, game type, academic major, and device used for gaming. However, there was a tendency toward configural non-invariance across groups of monthly income (ΔCFI = 0.022, ΔTLI = 0.020) and strict non-invariance across groups of gender and game type (ΔCFI = 0.042, 0.028; ΔTLI = 0.037, 0.020). However, ΔRMSEA was within the acceptable range in all tests, which supports invariance of the IGDS9-SF.

The IGDS9-SF expressed good fit in the Turkish, Australian, and American subsamples as well (Table 2). It was invariant at the configural, metric and scalar levels across participants from the four countries. However, strict invariance was not maintained (ΔCFI = 0.147, ΔTLI = 0.120, ΔRMSEA = 0.027) due to minor misspecifications in the covariances of a few of the items among participants from Turkey, Australia, and the USA (Figure 1). A Kruskal Wallis test revealed significant differences in the levels of IGD across countries (H (3) = 185.99, *p* = 0.001), with the lowest levels reported in participants from Turkey (median (IQR) = 13.0 (10.0–18.0)) and Sri Lanka (median (IQR) = 18.0 (13.0–22.0)) while the highest levels were reported in participants from the USA (median (IQR) = 21.0 (17.0–27.0)) and Australia (median (IQR) = 20.0 (16.0–25.0)). The Shapiro Wilk W test showed that the IGDS9-SF demonstrates a similar distribution in all the samples. Reliability tests showed that the internal consistency of the scale in the international subsamples ranged between very good and excellent (Table 2).

### 3.2. Reliability and Criterion Validity of the Internet Gaming Disorder Short-Form 9

Among Sri Lankan students, the IGDS9-SF expressed adequate internal consistency (coefficient alpha = 0.81), item-total correlations ranging between 0.341 and 0.611, and alpha if item deleted ranging between 0.779 and 0.816. It also demonstrated adequate criterion validity by exhibiting strong positive correlation with all motives of gaming measured by the MOGQ; the highest correlations were demonstrated with the dimensions of Escape and Coping (Table 3). It also correlated with the number of gaming hours, and the Kruskal Wallis test revealed significantly higher scores of the IGDS9-SF among those with daily gaming for more than three hours (H (2) = 88.8, *p* = 0.001). The IGDS9-SF did not correlate with the number of sleep hours (r = −0.051, *p* = 0.360), but it was negatively correlated with sleep quality (r = −0.120, *p* = 0.05). It negatively correlated with the single measure of self-esteem, albeit non-significantly.

The IGDS9-SF strongly correlated with Gender. The Mann Whitney U test revealed significantly higher IGD levels among males than females (U = 9261.0, z = −4.21, *p* = 0.001). Males recorded more daily time (>3 h) spent in gaming than females (χ^2^ (1) = 32.19, *p* = 0.001). The Mann Whitney U test also revealed significantly higher scores of Social, Escape, Coping, Fantasy, and Recreation motives among males than females (U = 9724.0, 10,934.5, 10,987.0, 10,619.0, 8382.0; z = −3.66, −2.19, −2.12, −2.59, −5.29; *p* = 0.001, 0.028, 0.034, 0.010, 0.001). IGD correlated with ethnicity, and the Mann Whitney U test revealed significantly higher IGD among Sinhala students compared with students from other ethnicities (U =7 446.0, z = −2.35, *p* = 0.019). 

Despite a tendency toward non-invariance across income groups, no significant differences between groups in the scores of the IGDS9-SF were noted among income groups (H (2) = 0.72, *p* = 0.696). Significant differences in IGD were recorded among game types, with the lowest scores reported among single offline players and the highest scores expressed among online multiplayers (H (2) = 9.70, *p* = 0.008). Multiple online players (28.0%) spent more than three hours gaming a day compared with 11.0% of offline players and 2.0% of single online players (χ^2^ (4) = 41.54, *p* = 0.001). They also exhibited significantly higher levels of Social, Competition, Skill development, and Recreation motives (H (2) = 47.49, 20.06, 11.83, 18.99; all *p* values = 0.001).

## 4. Discussion

Concerns about psychometric equivalence of IGD measures in different parts of the world represent a challenge for adequate identification of IGD in different cultural contexts [24]. The psychometric properties of the IGDS9-SF have been largely tested in English-speaking, European, and a small number of less developed countries. Accordingly, the current study complements existing knowledge by examining the psychometric properties of this scale among respondents from Sri Lanka and evaluating its measurement invariance across four international groups from distinct cultural backgrounds.

Consistent with previous studies, data obtained from the Sri Lankan and international samples expressed good fit of the single factor structure of the IGDS9-SF, with all items adequately loading on this factor (Table 2, Figure 1). Multigroup CFA revealed non-invariance of this measure at the configural, metric, and scalar levels across the international participants as well as across various groups in the Sri Lanka sample. However, the IGDS9-SF did not hold strict invariance in the international sample, and there was a tendency toward strict non-invariance across groups of gender and game type in the Sri Lankan sample. Obviously, slight improvements in the fit of the IGDS9-SF were attained by correlating the error terms of item 7 (deceiving others) and item 9 (jeopardizing relationships) in the Turkish sample. Among Turkish adolescents, excessive social media use is associated with an interplay between family life satisfaction and social connectedness [35]. Therefore, the interaction between items of deception and jeopardizing relationships may reflect on a subtle relation factor underlying IGD in the Turkish context. On the other hand, correlating the error of item 5 (loss of interest) with item 8 (escape) and the errors of item 2 (withdrawal—Irritability when reducing or stopping use) and item 3 (tolerance—need to spend more time gaming) improved the fit in the Australian and American samples. It seems that gaming is adopted as a measure to escape negative emotions (loss of interest) in the Australian and American contexts. Meanwhile, the mood-modifying effect that results from achieving status or progress in online gaming diminishes in persons with prolonged engagement [36]. Therefore, loss of interest in previous gaming activities may derive tolerance—gaming for a longer time to achieve the previous satisfactory effect. Accordingly, lack of satisfaction of gamers’ emotional needs may be related to irritability upon reducing/stopping (withdrawal). Cumulative knowledge shows variations in the internal and external events which shape the psychosocial background of IGD among individuals meeting the diagnostic criteria of IGD [37]. Thus, further investigations are needed to explore different dynamics underlying IGD in the cultures addressed in our study and whether they affect people’s responses to the items of the scale. 

Although the loadings of items 2 and 3 were strong in our American and Australian samples, they contributed to metric non-invariance in Australian respondents in a previous assessment of time invariance across Australian and American gamers. In the same Australian sample, items 4, 6, 8, and 9 were involved in partial scalar non-invariance of the scale [24]. Likewise, the single IGD factor was replicated in Polish gamers; however, poor fit was expressed by item 6 (continuation), item 7, and item 8 [17]. In another study comprising participants from the UK, USA, and India, cross-country variations were noted for items pertaining to preoccupation, tolerance, deception, escape, impairment in daily activities, and lack of control [21]. These findings generally suggest that the IGDS9-SF possibly involves a general component of high time and energy investment into IGD, in addition to the specific core components of IGD [38,39].

Configural and metric invariance of the IGDS9-SF across countries in our study indicates that IGD is similarly conceptualized by participants from poor countries (e.g., Sri Lanka), countries with evolving economies (e.g., Turkey), and developed countries (Australia and the USA). Strict non-invariance is rarely achieved while one third of the commonly used psychometric measures exhibit partial non-invariance [22,40]. Because the scale demonstrated scalar invariance across countries, the findings of this study show that the IGDS9-SF can be reliably used to compare IGD levels in those countries [40]. Significant differences in the level of IGD were noted among the international groups, with the highest occurrence reported among participants from the USA and Australia. On the contrary, Turkish students reported the lowest level followed by Sri Lankan students. In accordance, aggregate dates show that the levels of IGD and social media abuse are evidently higher in Europe and America than Asia [41]. These findings lend further support to previous studies reporting more pathologized IGD scores less close to the mean among respondents from individualistic cultural orientations such as the USA and close to normal scores among collectivistic or less individualistic countries (e.g., Turkey and the UK) [10,21]. It is worthy to note that the IGDS9-SF was examined in India before. However, it was administered in English [21], with a possibility that the scale may demonstrate different properties if it was administered among Indians who speak only local languages. In the present study, the IGD-SF9 was presented both in English and Sinhala, and it maintained perfect invariance across languages used, with no significant IGD differences between Sinhala and respondents from other ethnicities. Given the proximity as well as the cultural and geographical similarities between India and Sri Lanka, the IGDS9-SF seems to be a suitable measure in this region of south Asia.

Non-invariance of the IGDS9-SF across different groups (e.g., gender, ethnicities, and game types) denotes that the IGDS9-SF operates evenly as an IGD measure among youth from different social backgrounds in Sri Lanka. However, IGD significantly correlated with gender, with males expressing significantly higher IGD levels, IGD motives, and gaming time than females. This finding is consistent with reports of a recent meta-analysis comprising studies across 22 countries, which confirms that men exhibit significantly higher levels of IGD than women. On the other hand, women express higher levels of excessive social media abuse than men [41]. Increased risk for IGD among males is likely to be attributed to gender differences in the activation and connectivity of brain regions associated with the mesocorticolimbic reward system [37]. While females retreat to social networking to meet their need to relate [39], it seems that males retreat to gaming to satisfy their need to relate, as noted by significantly higher levels of Social motives among males in the present study. Moreover, online multiplayers were a majority in this study (Table 1); they also expressed the greatest time spent gaming, as well as the highest levels of IGD and gaming motives (Social, Competition, Skill development, and Recreation), particularly compared to offline single players. A qualitative investigation shows that distinctions between online and offline gaming can be largely shaped by the development of relationships, norms, and expectations—a person frequently plays with ’people like me’ [42]. Overall, males engage in gaming more than females, engage more in online multi-playing, and endorse more Social and Competition motives, denoting that gaming is probably employed to meet social needs.

## 5. Strength and Limitations

This study has the strength of integrating public data to examine the psychometric properties of the IGDS9-SF in Sri Lanka and its measurement invariance across four countries with different cultural backgrounds. However, the study entails numerous limitations that must be acknowledged. The use of public data in the analysis makes us unable to answer pivotal questions. For example, it is not clear how the IGDS9-SF was translated into Sinhala and whether it was back translated into English before data collection. Priori calculations of sample sizes were not performed. Details on the number of people initially contacted as well as the specific sampling method are not available. The cross-sectional design precluded the evaluation of test-retest reliability of the scale. Lack of sociodemographic characteristics in the international sample makes it impossible to examine invariance of the scale across specific groups (e.g., gender, age, and education). The reported differences in IGD across countries may not be sufficiently accurate. This is because participants in the American and Australian samples were recruited from online gaming platforms. On the other hand, gaming was self-reported by the Sri Lankan and Turkish participants—a possibility of reporting bias. Future studies may remedy the flaws implicated in the present study.

## 6. Conclusions

The IGDS9-SF expressed good fit and invariance across different groups, along with satisfactory levels of reliability and criterion validity in Sri Lankan students. Males and online multiplayers spent more time playing and expressed higher levels of IGD and playing motives. Significantly higher levels of the social dimension of the MOGQ in both groups suggest that gaming is used to meet the need to relate. Future investigations of relevant factors (e.g., family relations, social connectedness, social skills/competence) may provide further explanations of the use of gaming by males to satisfy their social needs.

In addition to demonstrating good fit of the unidimensional structure among the Turkish, Australian, and American subsamples, the IGDS9-SF also expressed configural, metric, and scalar invariance across countries. Therefore, this scale may be reliably used to compare IGD levels in those countries. The covariances of error terms of items related to deceiving others and jeopardizing relationships in the Turkish sample suggest the involvement of social factors in IGD in this culture. On the other hand, covariances between the error terms of items 5 and 8 as well as items 2 and 3 in the Australian and American subsamples indicate that gaming is employed by respondents from those countries to escape negative emotions. However, failure of gaming to resolve negative emotions may contribute to continued gaming (tolerance) and withdrawal symptoms upon sudden cessation. Future studies examining cultural invariance of the IGDS9-SF may need to identify the dynamics underling IGD in different countries and whether they may affect individual responses to the items of the IGDS9-SF.

## Figures and Tables

**Figure 1 healthcare-10-00490-f001:**
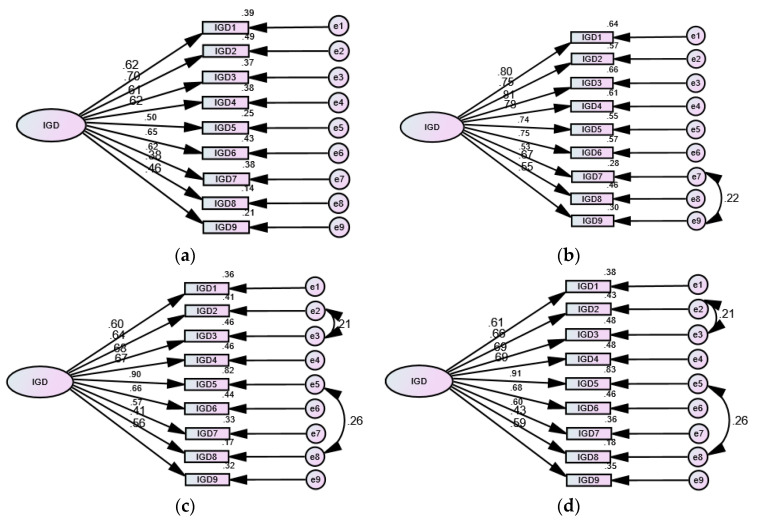
Factor structure of the Internet Gaming Disorder 9—Short Form (IGDS9-SF) among participants from Sri Lanka (**a**), Turkey (**b**), Australia (**c**), and the USA (**d**).

**Table 1 healthcare-10-00490-t001:** Sociodemographic, academic, and gaming characteristics of university students from Sri Lanka.

Participant Characteristics	(N = 322)
Age in years mean (SD)	17.2 ± 0.6
Gender	
Males	182 (56.5%)
Females	140 (43.5%)
Ethnicity	
Sinhala	249 (77.3%)
Others	73 (22.7%)
Language used	
Sinhala	152 (47.2%)
English	170 (52.8%)
Major	
Physical education	88 (27.3%)
Commerce	106 (32.9%)
Biology	67 (20.8%)
Arts	51 (15.9%)
Others	10 (3.1%)
Monthly income per household (SLR) ▲	
<100,000	124 (38.5%)
<200,000	106 (32.9%)
>200,000	92 (28.6%)
Gaming Type	
Offline single player	109 (33.9)
Online multiplayer	164 (50.9)
Online single player	49 (15.2)
Gaming hours/day	
One hour or less	172 (53.4)
Two to three hours	91 (28.3)
More than three hours	59 (18.3)
Device	
Mobile	206 (64.0%)
Others	116 (36.0%)
Sleep hours/day	
Five hours or less	102 (31.7%)
Six hours or more	220 (68.3.2%)

SLR: Sri Lankan rupee; ▲: one SLR is equal to $0.0049 or €0.0044.

**Table 2 healthcare-10-00490-t002:** Goodness-of-fit indices for the one-factor structure of the Internet Gaming Disorder Scale 9—Short form (IGDS9-SF) among university students from Sri Lanka, its invariance across countries, normality tests, and internal consistency.

Groups	Invariance Levels	χ^2^	df	*p*	Δχ^2^	Δdf	*p* (Δχ^2^)	CFI	ΔCFI	TLI	ΔTLI	RMSEA	ΔRMSEA	SRMR	W ◭	Coefficient Alpha
Countries	Sri Lanka	41.85	27	0.034				0.978		0.970		0.041		0.0357	0.961	0.811
	Turkey	73.68	26	0.001				0.967		0.954		0.076		0.0353	0.838	0.902
	Australia	109.95	25	0.001				0.965		0.950		0.068		0.0359	0.950	0.862
	USA	113.02	25	0.001				0.967		0.953		0.069		0.0350	0.965	0.876
	Configural	351.23	108	0.001				0.956		0.941		0.038		0.0357		
	Metric	459.81	132	0.001	108.578	24	0.001	0.940	0.016	0.935	0.006	0.040	−0.002	0.0577		
	Scalar	473.97	135	0.001	14.163	3	0.003	0.938	0.002	0.934	0.001	0.041	−0.001	0.0735		
	Strict	1307.43	162	0.001	833.458	27	0.001	0.791	0.147	0.814	0.120	0.068	0.027	0.1086		

χ^2^: chi-square; df: degrees of freedom; CFI: comparative fit index; TLI: Tucker–Lewis index; RMSEA: root mean square error of approximation; SRMR: standardized root mean residual; ◭: Shapiro–Wilk W test with all *p* values < 0.01; values in boldface indicate variance.

**Table 3 healthcare-10-00490-t003:** Descriptive statistics of and correlation of the Internet Gaming Disorder Scale 9—Short form (IGDS9-SF) with criterion variables among university students from Sri Lanka.

Variables	1	2	3	4	5	6	7	8	9	10	11	12
1. IGDS9-SF	-											
2. Gender	0.235 **	-										
3. Ethnicity	0.131 *	0.288 **	-									
4. Sleep quality	−0.120 *	0.082	0.000	-								
5. Gaming hours/day	0.532 **	0.312 **	0.170 **	−0.066	-							
6. Self-esteem	−0.104	0.084	0.030	0.193 **	-.008	-						
7. Social	0.514 **	0.204 **	−0.014	−0.038	0.463 **	0.033	-					
8. Escape	0.616 **	0.123 *	0.043	−0.161 **	0.383 **	−0.122 *	0.537 **	-				
9. Competition	0.367 **	0.085	0.028	−0.059	0.340 **	0.017	0.495 **	0.452 **	-			
10. Coping	0.513 **	0.119 *	−0.042	−0.134 *	0.295 **	−0.149 **	0.503 **	0.708 **	0.426 **	-		
11. Skill development	0.398 **	0.060	−0.098	−0.030	0.334 **	−0.057	0.552 **	0.538 **	0.534 **	0.590 **	-	
12. Fantasy	0.451 **	0.145 **	0.073	−0.074	0.321 **	−0.017	0.469 **	0.604 **	0.459 **	0.505 **	0.497 **	-
13. Recreation	0.389 **	0.295 **	0.098	0.020	0.448 **	−0.024	0.464 **	0.254 **	0.356 **	0.439 **	0.418 **	0.322 **
Median	18.0	-	-	7.0	-	4.0	8.0	7.0	9.0	9.0	8.0	7.0
IQR (Q1–Q3)	13.0–22.0	-	-	5.0–8.0	-	3.0–4.0	5.0–8.0	5.0–10.0	6.0–12.0	7.0–12.0	5.8–12.0	4.0–10.0

*: *p* < 0.05; **: *p* < 0.01.

## Data Availability

The datasets used to produce the current article are publicly available in Mendeley at: https://data.mendeley.com/datasets/8r2jgm6ygh/1 [26], https://data.mendeley.com/datasets/k698sznwf6/3 [28], and in DANS at: https://easy.dans.knaw.nl/ui/datasets/id/easy-dataset:162945 [29], (accessed on 2 August 2021).

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
