# Peer review of "The Nine-Item Internet Gaming Disorder Scale (IGDS9-SF): Its Psychometric Properties among Sri Lankan Students and Measurement Invariance across Sri Lanka, Turkey, Australia, and the USA"

_healthcare, 2022, doi:10.3390/healthcare10030490_

Round 1

Reviewer 1 Report

Review of article titled Measurement invariance of the Nine-Item Internet Gaming 2 Disorder Scale (IGDS9-SF) across Siri Lanka, Turkey, Australia, and the USA”.

  1. The topic of the article is an important one. Measurement invariance seems required in studies involving two or more nations. Thus, scale measurement invariance needs to be addressed in comparative, international, global studies. The scale measures a health construct that is both current and of practical relevance.
  2. Clarity of the article may improve if, the Sri Lanka study is developed first, and such is followed by the examination of the scale in the four countries studied.
  3. Considering the title of the article, it may be reasonable to expand, provide more detail, on the different types of invariances studied.
  4. Given the country differences found concerning IGD levels, it seems that scalar invariance was not found.
  5. Needs to be consistent with the terms used in the main body of the manuscript and those used in Table 2. Please use same terminology throughout the paper.
  6. Please indicates the meaning of acronyms (e.g., PUBG, DSM-5).
  7. The following statement, on page 2, “These nine criteria have been de-91 scribed based on existing research on pathological gambling and substance use disorder, 92 associating IGD measurement with methodological issues [8]” , may need to be edited to make it clearer.
  8. On line 26, “Given the widespread of IGD and related mental health and academic adverse effects” needs to read as follows “Given the widespread use of IGD and related mental health and academic adverse effects”.
  9. On Table 1, authors need to clarify if monthly income is per person or per household.
  10. On page 4, authors need to use consistently N or n.
  11. It may be preferable to avoid using etc.
  12. In section 2.2., authors need to provide information regarding translation and back-translation of questionnaire Sinhala-English.
  13. Table 2 refers CI but such data do not appear in the Table.
  14. The article may benefit from a separate section addressing limitations and future research.

Author Response

Dear Reviewer,

Greetings and thank you so much for your enormous help.

The response to the comments is in the attached file.

Best regards,

Reviewer 2 Report

This is an interesting manuscript, dealing with the cross-cultural validation of the IGDS9-SF. The authors exploited their experience on the validation of questionnaires and developed a really interesting study. The manuscript is well written.

I have however a couple of questions.

The most relevant concerns ethical issues. The authors state that "Data collection was obtained through an anonymous online survey conducted through SurveyGizmo, and an ethical approval for data collection was issued by the ethics committee of Cairnmillar Institute". They also declare that they considered the approval waived for datasets freely available. However, in the final notes, the authors declare that no ethical approval was obtained. Standard Good Research Practices require that a formal ethical approval is obtained if any procedure (be it an evaluation or a treatment) is applied to someone. If the available dataset were not obtained after an informed consent, they should not be used. This point requires therefore to be clarified.

I also wonder if a priori sample size was calculated and how or why not.

Author Response

(The authors gave the same response as above.)

Round 2

Reviewer 2 Report

I think the authors did a very good work with the actual version of their manuscript.

As far as I can understand, the study was conducted using data from previous studies, without collecting new data. If this is correct, the ethical issues are solved and the manuscript could be published as is 

If however new data were collected in any form for the present study (even if they are only demographics and questionnaires), I think that the paper cannot be published because it lacks ethical validation.

Author Response

Manuscript ID: healthcare-1611378

Authors’ response to the comments of Reviewer 2

Dear Reviewer 2,

Greetings!!

We are grateful for the decent and understanding approach of the reviewer.

Yes, the first statement indicated by the reviewer is 100% correct “As far as I can understand, the study was conducted using data from previous studies, without collecting new data. If this is correct, the ethical issues are solved and the manuscript could be published as is “. Accordingly, we have included the URL of all the datasets included in the analysis for the readers’ reference (line 442-444).

If however new data were collected in any form for the present study (even if they are only demographics and questionnaires), I think that the paper cannot be published because it lacks ethical validation.

For clarity, we have indicated in the text that no new data were collected apart from the used public data (line 438-439).

Thank you once again, and we hope that the manuscript has been satisfactorily modified and that the current version will be suitable for publication.

Best regards,
